# AuxVLA: Auxiliary Task Learning and Multi-Modal Enhancement for Vision-Language-Action Models in Mobile Manipulation

## Abstract

Vision-Language-Action (VLA) models have shown promise for robotic control, but their application to complex household manipulation tasks remains challenging. In this work, we propose AuxVLA, a comprehensive approach that enables VLA models to control mobile manipulation robots in domestic environments through both auxiliary task co-training and enhanced input modalities. Our method addresses the challenges of controlling high-dimensional action spaces (13 dimensions for both arm and mobile base) where direct imitation learning typically yields suboptimal results. AuxVLA incorporates two complementary strategies: (1) leveraging multi-view visual inputs and depth information to provide richer spatial context, and (2) co-training with auxiliary decoders that predict interpretable intermediate representations including global robot position, joint configurations, grasp affordances, target object relative positions, and segmentation masks from shared visual-language features. Through evaluation on home rearrangement tasks, AuxVLA demonstrates favorable performance across picking, placing, opening and closing tasks. We hypothesize that auxiliary supervision on interpretable representations enhances spatial understanding and scene reasoning capabilities, while enriched sensory inputs provide necessary spatial context for precise manipulation. These findings suggest that combining auxiliary objectives with multi-modal sensing offers a promising direction for VLA models in mobile manipulation, contributing to the development of more capable domestic robots.

## 1 Introduction

Vision-Language-Action (VLA) models, which leverage the powerful representations of pre-trained Vision-Language Models (VLMs) (Qwen et al., 2025; Karamcheti et al., 2024b), have emerged as a leading paradigm for translating natural language commands into robotic actions (Brohan et al., 2023a; Li et al., 2024a; Qu et al., 2025; Intelligence et al., 2025). These models have demonstrated remarkable success in learning a wide array of skills, setting new benchmarks for generalization and semantic understanding in robotic control (Mees et al., 2022; Li et al., 2024b; Black et al., 2024).

However, much of this recent success has been concentrated in constrained, table-top environments involving single or dual-arm manipulators Kim et al. (2024); Li et al. (2024a). While valuable, these settings do not capture the complexities of real-world domestic scenarios. Household tasks require robots to navigate through spaces, interact with articulated objects like doors and drawers, and reason about their own state relative to a dynamic, unstructured environment. This leap from static to mobile manipulation introduces significant challenges (Szot et al., 2022; Fu et al., 2024), including partial observability, high-dimensional continuous action spaces that fuse navigation and manipulation, and a greater need for robust spatial and scene-level reasoning.

Despite the growing interest in applying VLA models to mobile manipulation, this remains a relatively nascent area of research. Current approaches to household robotics have primarily relied on modular systems that decompose tasks into separate navigation and manipulation components (Yarats et al., 2021; Lu et al., 2025; Cheng et al., 2025), or end-to-end learning methods that struggle with the complexity of unified control (Fu et al., 2024; Chen et al., 2025). The application

of VLA models to mobile manipulation represents an emerging paradigm that seeks to harness the rich semantic understanding and reasoning capabilities of large-scale vision-language pre-training for these challenging scenarios. However, existing VLA architectures, when directly applied to mobile manipulation tasks, exhibit substantial performance limitations. We observe that standard VLA models trained through direct imitation learning achieve only modest success rates on household tasks, suggesting that current approaches fail to adequately leverage the sophisticated reasoning abilities inherent in pre-trained VLMs.

In this work, we investigate how to bridge this gap and explore the applicability of VLA models to complex mobile manipulation tasks in household settings. We hypothesize that the standard approach of direct imitation learning—predicting a high-dimensional action vector from a visual-language representation—provides insufficient supervisory signal for the model to learn the rich, multi-faceted understanding required for these tasks. To address this, we introduce **AuxVLA**, a comprehensive framework that enhances the learning process through two primary strategies: enriching the model's sensory input and providing denser learning signals through auxiliary co-training.

Our work makes the following contributions:

- We propose an efficient co-training strategy that leverages a shared visual-language backbone to simultaneously predict actions and a suite of auxiliary tasks, including the robot's global position, joint configuration, grasp state, and object-centric properties.

- We demonstrate that these auxiliary objectives act as a powerful form of explicit supervision, forcing the model to learn more interpretable and spatially aware representations from its latent features.

- We systematically explore how different input modalities, specifically multi-view imagery and depth information, can provide richer spatial context to improve VLA performance in mobile manipulation.

- We validate our approach on the challenging ManiSkill-HAB benchmark (Shukla et al., 2025), showing that AuxVLA achieves average success rates of 73% compared to 60% for direct imitation learning on home rearrangement tasks, demonstrating the effectiveness of our proposed enhancements on a common VLA architecture.

These findings suggest that enriching both the inputs and the training objectives is a critical and promising direction for scaling VLA models to the complexities of real-world domestic robotics.

## 2 RELATED WORK

**Vision-Language-Action models.** Vision-Language-Action (VLA) models have emerged as a dominant paradigm for robotic control, leveraging large-scale pretrained models to translate multimodal prompts into actions. Pioneering works such as Shridhar et al. (2021); Reed et al. (2022); Brohan et al. (2023b); Liu et al. (2024); Zhang et al. (2024); Qu et al. (2025); Zhang et al. (2025) demonstrated that end-to-end training on large datasets (O'Neill et al., 2024; Deng et al., 2025) could produce highly capable policies. Some models discretize the continuous action space (Brohan et al., 2023a; Kim et al., 2024; Belkhale & Sadigh, 2024), enabling the VLM backbone (Karamcheti et al., 2024a) to directly predict action tokens in an autoregressive fashion. Others append a specialized action expert (Black et al., 2024; Li et al., 2024a; Yating Wang, 2025), which generates continuous actions from the VLM's latent features. Despite their success, the capabilities of these VLA frameworks are predominantly demonstrated on tabletop manipulation tasks, leaving the challenges of mobile manipulation largely unaddressed.

**Mobile Manipulation.** Solving mobile manipulation requires the tight synergy of locomotion and arm control to carry out human instructions. One dominant strategy employs a hierarchical system where a Vision-Language Model (VLM) (Achiam et al., 2023) serves as a high-level planner (Wake et al., 2024), decomposing instructions into simpler subtasks. These subtasks are then handled by separate, specialized policies for navigation and manipulation (OpenAI et al., 2019; Haarnoja et al., 2018; Joshi et al., 2020; Fu et al., 2024; Jiang et al., 2025; Cheng et al., 2025). This modularity, however, isolates the foundation model's rich, pre-trained knowledge from the final motor control, limiting its direct influence on physical interaction.

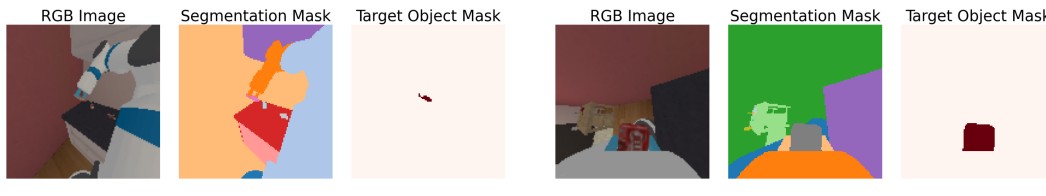

(a) Head camera view                    (b) Hand camera view

Figure 1: **Multi-view segmentation data used for auxiliary task training.** Each group shows (left) the original RGB observation, (middle) segmentation masks for all objects in the scene with different colors representing different object instances, and (right) the processed binary mask highlighting only the target object of interest. (a) Head camera view provides a global perspective of the manipulation scene. (b) Hand camera view offers a close-up perspective focused on the manipulation area. The target object masks are derived from the full segmentation by identifying the target object ID and creating binary masks to focus the model's attention on the relevant manipulation target.

## 3 DATASET

We adopt the data generation pipeline from ManiSkill-HAB (Shukla et al., 2025), a benchmark for low-level manipulation in home rearrangement tasks, and extend it to generate demonstration data enriched with auxiliary task annotations. The dataset comprises simulation-based trajectories across three long-horizon household tasks: TidyHouse, PrepareGroceries, and SetTable, totaling 44K episodes with 8.2M transitions.

**Task Composition.** Each long-horizon task is decomposed into 4 fundamental manipulation subtasks: **Pick**, **Place**, **Open**, and **Close**, with varying difficulty levels. TidyHouse involves pick-and-place operations with medium to hard difficulty objects and receptacles. PrepareGroceries focuses on hard-difficulty pick-and-place scenarios requiring precise manipulation. SetTable includes all 4 subtasks, spanning easy to medium difficulty levels. Pick and place tasks require fine-grained manipulation control for accurate grasping and positioning, whereas open and close tasks emphasize broader locomotion and approach strategies. The diversity in task complexity and object types provides comprehensive coverage of household manipulation scenarios.

**Auxiliary Task Data Distribution.** We strategically distribute auxiliary task training based on task relevance and data availability. For auxiliary tasks involving **global position, grasp state, and joint position (qpos) prediction**, we utilize the complete SetTable dataset (8K episodes, 1.6M transitions), which provides comprehensive robot state information across diverse manipulation scenarios. For **segmentation masks and object pose estimation**, we leverage pick-and-place data from all three tasks (40K episodes, 8M transitions), as these auxiliary tasks are only meaningful in scenarios involving target object manipulation and require clear object identification. Figure 1 shows an example of the segmentation data. This task-specific data allocation ensures that each auxiliary decoder receives relevant and sufficient training examples while maximizing the utilization of our diverse demonstration dataset.

## 4 MODEL

### 4.1 OVERALL ARCHITECTURE

AuxVLA is a lightweight 1.3B parameter model consisting of a pre-trained Prismatic VLM (Karamcheti et al., 2024a) backbone and an optional action expert, designed to efficiently handle multimodal inputs for robotic control tasks. The VLM backbone comprises three key components: (1) a **visual encoder** that fuses complementary features from DINOv2 (Oquab et al., 2024) (improving spatial understanding) and SigLIP (Zhai et al., 2023) (providing rich semantic representations) following dual-encoder approach in Kim et al. (2024), enabling the model to process both RGB and depth inputs from multiple camera viewpoints; (2) a **large language model backbone** (Qwen2.5-0.5B (Qwen et al., 2025)) that serves as the central reasoning component, processing textual instructions and integrating multi-modal information for decision making; and (3) a **trainable projector** that maps high-dimensional vision features to the language embedding space, allowing seamless fusion of visual and linguistic representations within the transformer architecture.

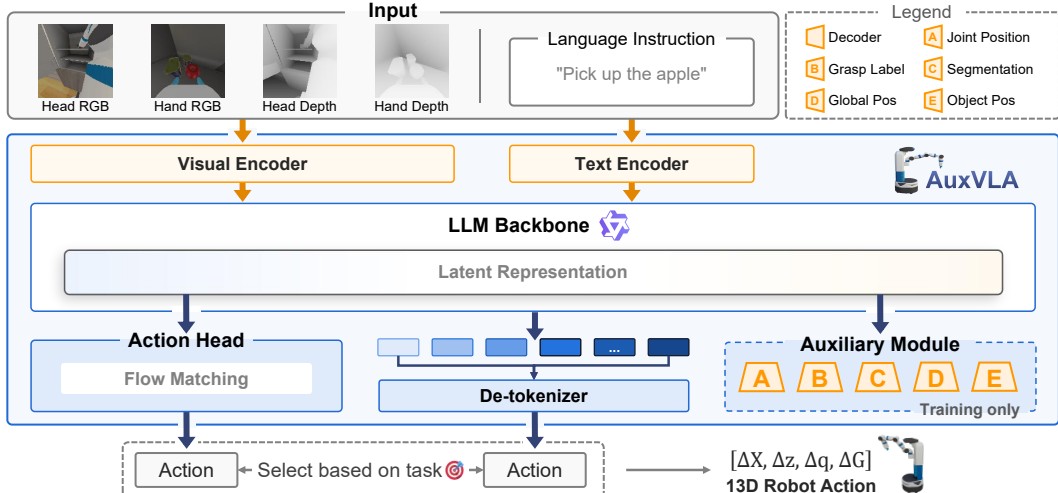

Figure 2: **AuxVLA Architecture Overview.** The model processes multi-modal inputs, including RGB images and normalized depth maps from head and hand cameras, alongside a natural language instruction. During training time, the latent representation from LLM is then passed to the decoders for auxiliary task predictions. Finally, the model predicts a 13-dimensional action vector, generated either directly by the LLM backbone or through a specialized Flow Matching action expert depending on task type. The action vector consists of the following: $\Delta X$, a 3D vector representing the **base's pose (position+orientation)**; $\Delta z$, a 1D scalar representing the change in the **torso's height**; $\Delta q$, a 7D vector representing the change in the **arm's joint angles**; $\Delta G$, a 2D vector representing the change in the **gripper's state** (one for each finger).

Additionally, we incorporate an optional 100M parameters **Flow Matching** (Lipman et al., 2023) **action expert** following $\pi_0$ (Black et al., 2024), which generates continuous, high-precision actions conditioned on the rich latent representations extracted from the VLM backbone. This dual-prediction architecture provides flexibility in action generation: the model can either directly predict discrete action tokens, or leverage the action expert to generate continuous actions. The choice between these prediction modes can be adapted based on specific task requirements. The complete architecture, illustrating the flow from multi-modal inputs through the VLM backbone to auxiliary decoders and action prediction, is shown in Figure 2.

## 4.2 INPUT MODALITIES

To enhance the spatial understanding and situational awareness of our VLA model, we systematically investigate the impact of different input modalities beyond standard single-view RGB observations. Specifically, we explore three key modality enhancements: (1) **Multi-view RGB inputs** that incorporate both head-mounted and hand-mounted camera observations, providing complementary perspectives of the manipulation scene compared to relying solely on head observations; (2) **Depth information** from head and hand cameras and normalized through Eq. 1, which offers explicit geometric understanding of object distances and spatial relationships in the environment; and (3) **Temporal context** through historical observations from the past 4 timesteps, enabling the model to leverage sequential information for better action planning. We conduct comprehensive ablation studies to evaluate the individual and combined effects of these modality additions on task performance. Our experiments reveal that multi-view RGB observations combined with their corresponding depth images yields the best performance, significantly improving the model's ability to understand complex spatial relationships and object interactions in household manipulation tasks. The detailed ablation results and analysis of each modality's contribution are presented in Section 6.

$$p_{obs} = 1 - \tanh\left(\frac{\text{depth value}}{1000}\right) \tag{1}$$

### 4.3 DECODERS

We design auxiliary decoders that operate on the VLM backbone's latent representations to enable multi-task learning. The decoders reconstruct different aspects of the current state, providing complementary supervision signals for various aspects of robotic manipulation. Structure details are listed in Appendix A.2.

**MLP-based Decoders.** We implement three regression and classification decoders using similar MLP architectures with progressive dimensionality reduction. The **Global Position Decoder** predicts the robot's 2D coordinates (x, y), the **Grasp Success Decoder** performs binary classification to determine grasping state, and the **Object Pose Decoder** predicts 7-dimensional object poses (3D position + quaternion orientation). Their respective loss functions are defined in Eq. 2, where $\hat{\mathbf{p}}$ and $\mathbf{p}$ are predicted and ground truth global positions, $y$ and $\hat{y}$ are ground truth and predicted grasp labels, $\mathbf{t}$ represents 3D position, and $\mathbf{q}$ represents quaternion orientation.

**Transformer-based QPos Decoder.** For the 12-dimensional joint configuration prediction, we employ a Vision Transformer architecture with learnable mask tokens that attend to VLM features through multi-head self-attention. This design captures complex joint dependencies while leveraging spatial understanding from VLM features, using fixed sine-cosine positional encodings and MSE loss for joint angle regression as shown in Eq. 3, where $\hat{\mathbf{J}}$ and $\mathbf{J}$ are predicted and ground truth 12-dimensional joint configurations.

**CNN-based Mask Decoder.** The segmentation decoder generates 128×128 binary masks for target objects using an efficient CNN upsampling pathway. Starting from 8×8 spatial resolution, it progressively upsamples through transposed convolutions with GELU activations and batch normalization, optimized for large-scale training on our dataset using binary cross-entropy loss as shown in Eq. 3, where $\hat{\mathbf{M}}$ and $\mathbf{M}$ are predicted and ground truth segmentation masks.

$$\mathcal{L}_{pos} = \text{MSE}(\hat{\mathbf{p}}, \mathbf{p}), \quad \mathcal{L}_{obj} = \|\hat{\mathbf{t}} - \mathbf{t}\|_2^2 + (1 - |\hat{\mathbf{q}} \cdot \mathbf{q}|), \quad \mathcal{L}_{grasp} = \text{CrossEntropy}(\hat{y}, y) \quad (2)$$

$$\mathcal{L}_{qpos} = \text{MSE}(\hat{\mathbf{J}}, \mathbf{J}), \quad \mathcal{L}_{seg} = \text{CrossEntropy}(\hat{\mathbf{M}}, \mathbf{M}) \quad (3)$$

**Multi-task Loss Function.** The total training loss combines the main action prediction loss with weighted auxiliary losses:

$$\mathcal{L}_{auxiliary} = \lambda_{pos}\mathcal{L}_{pos} + \lambda_{grasp}\mathcal{L}_{grasp} + \lambda_{qpos}\mathcal{L}_{qpos} + \lambda_{obj}\mathcal{L}_{obj} + \lambda_{seg}\mathcal{L}_{seg} \quad (4)$$

where the $\lambda$s are task-specific weighting coefficients tuned to balance contributions across diverse auxiliary objectives. Each decoder operates independently on shared VLM representations, enabling simultaneous learning of complementary manipulation aspects.

## 5 TRAINING SCHEME

We adopt a progressive multi-stage training approach to effectively integrate auxiliary decoders and flow-matching action head with the pre-trained VLM backbone while preserving the model's foundational capabilities.

### 5.1 PRELIMINARY EXPERIMENT

Our preliminary experiments revealed that directly co-training randomly initialized auxiliary decoders with the VLM backbone led to performance degradation. This phenomenon occurs because the untrained decoders generate large, noisy gradients that interfere with the learned representations in the pre-trained VLM.

### 5.2 TWO-STAGE PROGRESSIVE TRAINING

To address this challenge, we implement a two-stage training strategy that balances auxiliary task learning with preservation of VLM representations:

**Stage 1: Decoder Adaptation.** We freeze the gradient flow from auxiliary decoders to the VLM backbone, allowing only the discrete action token prediction path from the VLM to update the backbone parameters. During this phase, the randomly initialized decoders learn to interpret and

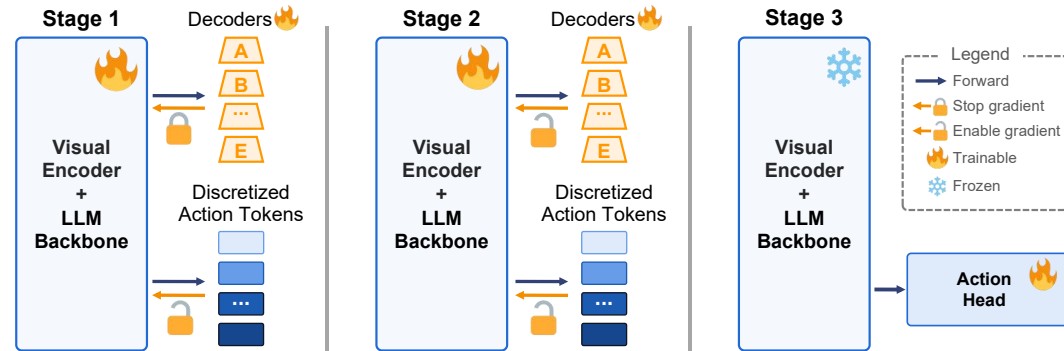

Figure 3: **Multi-stage Training Scheme.** Stage 1: Decoder adaptation phase where auxiliary decoders are trained while gradient flow to the VLM backbone is blocked, allowing decoders to learn from fixed VLM representations. Stage 2: Joint refinement phase with full gradient flow enabled, co-training all auxiliary decoders with the VLM backbone. Stage 3: Action head training phase where the VLM backbone is frozen to train the flow matching action head in isolation.

utilize the fixed latent representations from the VLM without disrupting the backbone's pre-trained knowledge, while the VLM continues learning to predict discrete action tokens for robot control. This stage enables the auxiliary decoders to adapt to the VLM's representational space and establish reasonable baseline performance on their respective tasks, while maintaining the VLM's core action prediction capabilities through discrete token generation.

**Stage 2: Joint Refinement.** After the decoders have stabilized, we enable full gradient flow, allowing auxiliary task losses to backpropagate through the entire network. In this phase, the auxiliary objectives guide the refinement of VLM representations to better capture manipulation-relevant features such as spatial relationships, object properties, and robot state information. The combined supervision from multiple tasks encourages the backbone to learn more comprehensive and robust representations that benefit the primary action prediction objective. This progressive approach ensures that auxiliary tasks enhance rather than hinder the model's learning process, as demonstrated by our ablation studies in Section 6.

### 5.3 Training the Action Head

Beyond the two-stage progressive training for auxiliary decoders, we introduce an additional training phase specifically for the optional flow matching action head. Our initial experiments revealed that training the flow matching action head alongside the VLM backbone, leads to optimization difficulties. Despite preventing the flow matching gradients from affecting the VLM parameters, the denoising loss still fails to converge effectively when trained concurrently with other objectives.

**Stage 3: Isolated Action Head Training.** To address this challenge, we implement a third training stage where we completely freeze all parameters of the pre-trained VLM backbone and train only the flow matching action head. In this stage, the action head learns to generate continuous 13-dimensional actions by denoising from the rich, frozen latent representations provided by the VLM backbone. This complete isolation allows the flow matching objective to converge properly without any interference from concurrent training processes, enabling the action head to develop robust action generation capabilities.

## 6 Experiments

### 6.1 Implementation Details

Given that VLA models for mobile manipulation remains a nascent research direction, we acknowledge that established baselines for these complex household tasks are relatively limited. Our experimental setup therefore focuses on demonstrating the effectiveness of our proposed auxiliary training strategies against a straightforward baseline implementation of direct imitation learning on the same architecture.

| Method | Pick | Place | Open | | Close | | |
| | All Obj. | All Obj. | Fridge | Drawer | Fridge | Drawer | Avg. |
|---|---|---|---|---|---|---|---|
| OpenVLA (Kim et al., 2024) | 0.00 | 0.19 | 0.02 | 0.00 | 0.00 | 0.04 | 0.04 |
| + Multiview | 0.06 | 0.35 | 0.14 | 0.38 | 0.00 | 0.53 | 0.24 |
| + Multiview + Depth | 0.12 | 0.41 | 0.43 | 0.30 | 0.00 | 0.67 | 0.32 |
| Base VLM + Multiview | 0.06 | 0.53 | 0.60 | 0.30 | 0.63 | 0.93 | 0.52 |
| + Multiview + Depth | **0.16** | **0.56** | **0.67** | 0.36 | **0.83** | **1.00** | **0.60** |
| + Multiview + Depth + History | 0.00 | 0.47 | 0.57 | **0.40** | 0.47 | **1.00** | 0.49 |

Table 1: Performance comparison between models taking different modalities of inputs. The Base VLM (Belkhale & Sadigh, 2024) uses DINOv2 + SigLIP as dual visual encoder and Qwen2.5-0.5B as LLM backbone. Reported numbers are success rates for each task. The best results are **bolded**.

All models are implemented in PyTorch and trained on 8 NVIDIA A100 GPUs. We use an Adam optimizer with constant learning rate $2e^{-5}$. Global batch size is set to 512. For the weights of losses in Eq. 4, we set $\lambda_{pos} = 1.0$, $\lambda_{grasp} = 5.0$, $\lambda_{qpos} = 1.0$, $\lambda_{obj} = 1.0$, $\lambda_{seg} = 1.0$. The models with segmentation decoders are train on pick and place subtasks for 5 epochs, while the others are train on all 6 subtasks for 10 epochs. We evaluate all models using the evaluation pipeline from ManiSkill-HAB (Shukla et al., 2025). For each task, we run 30 episodes and calculate the mean success rate. Sample visualization episodes can be found in Appendix A.1.

## 6.2 INPUT MODALITIES

Table 1 demonstrates the significant impact of enhanced input modalities on VLA model performance across different household manipulation tasks. All the models in this table are train on **SetTable** dataset for 10 epochs. The results reveal several key insights about the importance of multi-modal sensory information for robotic control.

**Baseline.** The original OpenVLA model ($\sim$7B parameters) (Kim et al., 2024) with single-view RGB input achieves extremely poor performance across all tasks (0.04 average success rate) after trained with same number of epochs as our models, highlighting the limitations of standard vision-language models when applied directly to complex household manipulation scenarios. This baseline establishes the critical need for enhanced sensory inputs in domestic robotics applications.

**Multi-view and Depth Benefits.** The addition of multi-view observations and depth information to the OpenVLA baseline demonstrates substantial improvements across all task categories. OpenVLA with enhanced modalities (multi-view + depth) achieves a dramatic $8\times$ improvement in average success rate (from 0.04 to 0.32), with particularly notable gains in drawer manipulation tasks where success rates increase from near-zero to 0.30-0.67. This improvement validates the importance of richer sensory information for spatial reasoning in household manipulation.

**Modality Ablation Analysis.** Performance is further enhanced when replacing the LLM backbone with the more efficient Qwen2.5-0.5B (Belkhale & Sadigh, 2024). Despite being smaller than the original OpenVLA model, the Qwen-based model achieves superior results. The model with multi-view and depth inputs reaches 0.60 average success rate, nearly doubling the performance of the enhanced OpenVLA variant. The systematic ablation study on the Qwen-based model reveals the individual contributions of each modality enhancement. Multi-view inputs alone provide substantial gains over single-view baselines, achieving 0.52 average success rate. Adding depth information further improves performance to 0.60, representing a 15% relative improvement and highlighting the value of explicit geometric information for manipulation tasks. However, incorporating temporal history (past 4 actions) degrades performance to 0.49. This counter-intuitive result may indicate that the model struggles to effectively integrate temporal information, or that the evaluated tasks are sufficiently reactive that historical context provides limited additional information.

## 6.3 AUXILIARY TASKS AND PROGRESSIVE TRAINING

Table 2 provides strong empirical evidence for the necessity of our progressive training approach when incorporating auxiliary decoders. All the models in this table are trained on **SetTable** subset for 10 epochs, with 3 epochs for stage 1 and 7 epochs for stage 2. The contrast between naive co-training (top section) and progressive training (bottom section) demonstrates the importance of

| Progressive training | Method | Pick All Obj. | Place All Obj. | Open Fridge | Open Drawer | Close Fridge | Close Drawer | Avg. S.R. |
|---|---|---|---|---|---|---|---|---|
| No | AuxVLA | 0.16 | 0.56 | 0.67 | 0.36 | 0.83 | 1.00 | 0.60 |
| | + all | 0.03 | 0.50 | 0.60 | 0.23 | 0.67 | 1.00 | 0.51 |
| Yes | + is_grasped | **0.30** | 0.53 | 0.83 | 0.57 | 0.80 | 0.93 | 0.66 |
| | + qpos | 0.23 | 0.67 | 0.87 | 0.70 | 0.90 | 0.90 | 0.71 |
| | + global pos | 0.07 | 0.27 | **0.90** | 0.70 | **0.97** | **1.00** | 0.65 |
| | + all | 0.13 | **0.70** | 0.87 | **0.77** | 0.90 | **1.00** | **0.73** |

Table 2: Ablation study on co-training with each auxiliary task. All the models are train on SetTable subset. "+all" here includes "is_grasped + qpos + global pos." The best results are **bolded** and the second-best results are underlined.

| | SetTable | | PrepareGrocery | | TidyHouse | | |
|---|---|---|---|---|---|---|---|
| Method | Pick | Place | Pick | Place | Pick | Place | Avg. |
| AuxVLA | 0.20 | 0.50 | 0.10 | 0.33 | 0.07 | 0.40 | 0.27 |
| + seg + obj_pos | **0.26** | **0.78** | **0.13** | **0.60** | **0.33** | **0.73** | **0.47** |

Table 3: Performance comparison between models trained with and without Segmentation and Object Position reconstruction task. The best results are **bolded**.

proper training methodology for multi-task learning in VLA models. Table 3 demonstrates the significant impact of incorporating segmentation masks and object position prediction as auxiliary tasks. Models in this table are trained on all **Pick** and **Place** data from **SetTable**, **PrepareGroceries** and **TidyHouse** tasks for 6 epochs, with 2 for stage 1 and 4 for stage 2.

**Failure of Naive Co-training.** When all auxiliary decoders are trained simultaneously with the VLM backbone from initialization ("AuxVLA + all" without progressive training in Table 2), performance degrades significantly across most tasks, dropping from 0.60 to 0.51 average success rate. This 15% performance degradation confirms our hypothesis that randomly initialized auxiliary decoders generate disruptive gradients that interfere with the pre-trained VLM representations.

**Progressive Training.** The progressive training approach not only recovers the baseline performance but substantially improves it. Training all auxiliary tasks with progressive methodology ("+ all" with progressive training in Table 2) achieves 0.73 average success rate, representing a 22% improvement over the best input modality configuration alone. This validates our two-stage training strategy where decoders first adapt to VLM representations before jointly refining the backbone.

**Individual Auxiliary Task Analysis.** The ablation study results in Table 2 and 3 reveals varying contributions from different auxiliary tasks.

- **Joint position (qpos):** Reconstructing qpos of current state provides the most substantial and consistent improvements across all task categories (0.71 average), particularly excelling in manipulation tasks like place (0.67) and drawer operations (0.70 and 0.90). This suggests that explicit joint awareness significantly enhances the model's understanding of manipulation dynamics.

- **Grasp label (is_grasped):** Grasp label prediction offers moderate but reliable improvements (0.66 average), with notable gains in pick tasks (0.30). This indicates that predicting grasp label help model gain better manipulation state awareness.

- **Global position (global pos):** Global position reconstruction shows more task-specific benefits, dramatically improving drawer and fridge opening tasks (0.90 and 1.00) while struggling with pick-and-place operations. This indicates its particular value for navigation-heavy scenarios.

- **Segmentation+object position (seg+obj_pos):** Unlike previous auxiliary tasks that showed selective improvements, segmentation and object position reconstruction provide consistent benefits across all manipulation scenarios. **Place tasks show the most substantial gains**, with improvements ranging from 55-81% across different environments. This

| | Pick | Place | Open | | Close | | |
|---|---|---|---|---|---|---|---|
| Method | All Obj. | All Obj. | Fridge | Drawer | Fridge | Drawer | Avg. |
| AuxVLA | 0.13 | 0.70 | **0.87** | **0.77** | **0.90** | **1.00** | **0.73** |
| AuxVLA + action head | **0.27** | **0.80** | 0.76 | 0.60 | 0.76 | 0.97 | 0.69 |

Table 4: Performance comparison between AuxVLA trained with and without flow matching action head. The best results are **bolded**.

pattern aligns with the intuitive importance of precise object localization and scene understanding for successful placement operations. **Pick tasks demonstrate more modest but consistent and notable improvements**. The relatively smaller improvements in pick tasks may reflect the continued challenge of grasp planning, which requires additional skills beyond object detection and localization.

**Synergistic Effects.** The combined auxiliary training ("+ all") achieves performance that generally matches or exceeds the best individual auxiliary task across most categories, with the overall average (0.73) representing near-optimal performance. This suggests that the auxiliary tasks provide complementary rather than redundant information, with each contributing unique aspects of spatial and manipulation understanding to the overall model capability.

## 6.4 ACTION HEAD

The action head is trained to predict action chunk of size 8, among which the first 2 actions are executed. Number of denoising step set to 10. Table 4 reveals the mixed effects of incorporating the flow matching action head with the overall average performance decreases from 0.73 to 0.69, indicating that the action head's benefits are selective and come with trade-offs.

**Task-Specific Performance Patterns.** The flow matching action head shows a clear dichotomy in its effectiveness across different manipulation primitives. **Pick** tasks benefit substantially from continuous action generation, with success rates more than doubling from 0.13 to 0.27. This improvement suggests that the fine-grained control afforded by continuous actions is particularly valuable for precise grasping motions, where small variations in approach angle, grip force, or contact points can significantly impact success. **Place** tasks also show moderate improvements (0.70 to 0.80), indicating that continuous control helps with the precise positioning required for object placement. Conversely, the action head causes notable performance drops in **Open** tasks across both fridge and drawer categories. Fridge opening decreases from 0.87 to 0.76, while drawer opening drops from 0.77 to 0.60. These results suggest that the flow matching action head excels at generating fine-grained manipulation actions but struggles with mobility-oriented control, where discrete action tokens may be more suitable for coordinated base movement and navigation.

**Implications for Task-Adaptive Control.** The mixed results strongly support our design choice to implement task-adaptive action generation, where the model can flexibly choose between discrete tokens and continuous actions based on task requirements. This flexibility allows AuxVLA to leverage the precision of continuous control for manipulation-heavy tasks while maintaining the decisiveness of discrete actions for mechanism operation tasks.

## 7 CONCLUSION AND FUTURE DIRECTIONS

This work demonstrates that VLA models can be effectively adapted for mobile manipulation through auxiliary task co-training and enhanced input modalities. AuxVLA achieves substantial improvements by incorporating multi-view depth inputs and auxiliary decoders, with progressive training proving essential for multi-task learning. These results establish that scaling VLA models beyond tabletop scenarios requires fundamental architectural and training enhancements. However, the primary limitation is the simulation-to-real gap, as experiments are conducted entirely in simulation. Future work should focus on real-world validation, extending to long-horizon household tasks, and exploring broader auxiliary tasks to guide strategic data collection for deployment. Additionally, exploring broader auxiliary tasks such as physics understanding and affordance prediction could improve performance while guiding strategic data collection for real-world deployment.

ETHICS STATEMENT

This work adheres to the ICLR Code of Ethics. Our research contributes to society and human well-being by advancing household robotics capabilities that can assist with domestic tasks. The simulation-based experiments avoid immediate harm to humans or the environment while developing technology intended for beneficial applications. We maintain scientific excellence through transparent reporting of methods, honest presentation of results including limitations, and acknowledgment of all contributions. Our work respects privacy by using only publicly available datasets without personal information. We have considered potential negative consequences and designed our system for constructive household assistance rather than harmful applications. Future real-world deployment should include appropriate safety measures and consideration of broader societal impacts.

REPRODUCIBILITY STATEMENT

To ensure reproducibility, we provide comprehensive implementation details throughout the paper. Our experiments are conducted on the publicly available ManiSkill-HAB benchmark, with specific data splits and preprocessing steps detailed in Section 3. The model architecture, including all auxiliary decoder specifications, is fully described in Section 4 and Appendix A.2. Our progressive training methodology is explicitly outlined in Section 5, including the specific epoch divisions and gradient flow configurations. All experimental hyperparameters, loss weights, and training configurations are provided in Section 6. The base VLM backbone (Qwen2.5-0.5B) and vision encoders (DINOv2, SigLIP) are publicly available pre-trained models. Code for reproducing our results, including data loading, model training, and evaluation scripts, will be made available upon publication.

THE USE OF LARGE LANGUAGE MODELS

Large language models were used as a general-purpose assist tool to aid in writing and polishing portions of this paper. Specifically, LLMs were employed to help refine sentence structure, improve clarity of technical explanations, and enhance the overall readability of the manuscript. All technical content, experimental design, results, and scientific conclusions remain the original work of the authors. The authors take full responsibility for all content in this paper, including any sections that were refined with LLM assistance.

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

# A   APPENDIX

## A.1   VISUALIZATION OF ROBOT EXECUTION IN MANISKILL-HAB

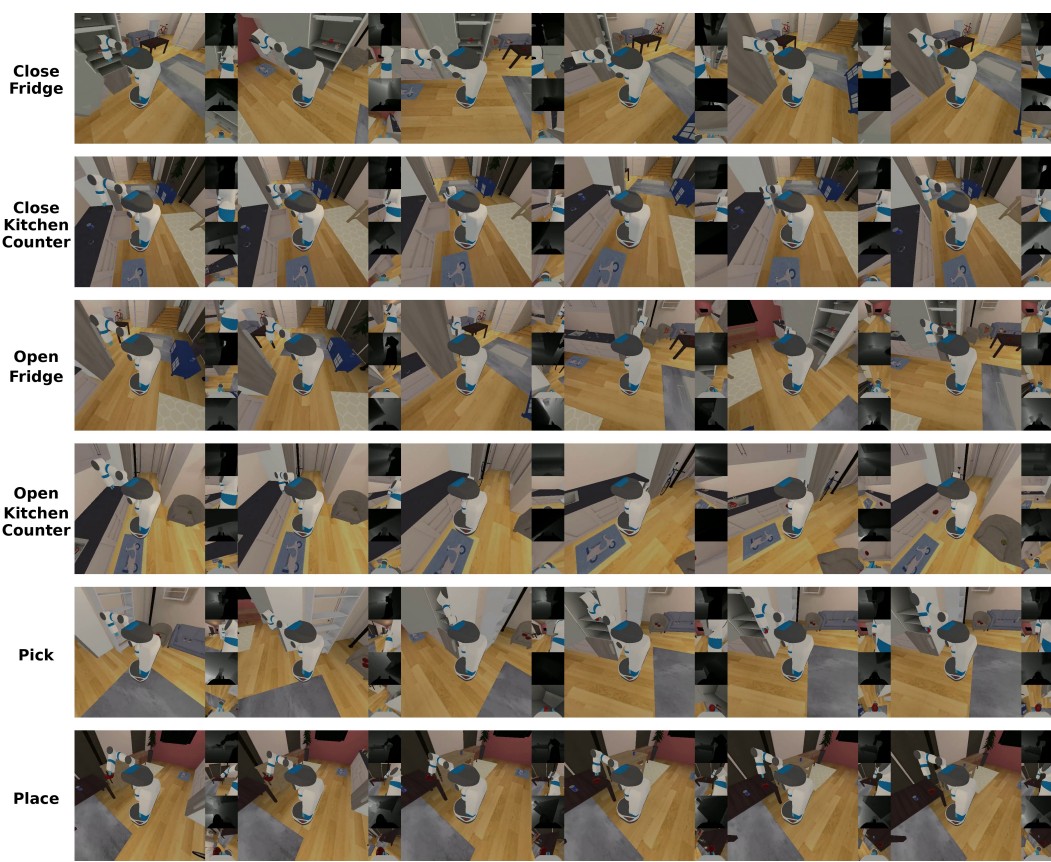

Figure 4: **Sample execution trajectories for six household manipulation tasks in ManiSkill-HAB evaluation**. Each row shows a temporal sequence of frames for a different task performed by AuxVLA. For each timestep, the large image shows the third-person view of the robot and environment, while the four smaller inset images display the multi-modal sensory inputs used by the model: head depth, head RGB, hand depth, and hand RGB observations (arranged vertically from top to bottom).

## A.2 DECODER SPECIFICATIONS

| Decoder | Architecture | Input | Output | Key Components |
|---|---|---|---|---|
| Global Pose Decoder | MLP | [B, 8, 896] | [B, 2] | Feature proj (896→512)
+ 3-layer MLP (512→256→128→2)
+ Global avg pooling |
| Grasp Success Decoder | MLP | [B, 8, 896] | [B, 1] | Feature proj (896→512)
+ 3-layer MLP (512→256→128→1)
+ Global avg pooling |
| QPos Decoder | Transformer | [B, 8, 896] | [B, 12] | Feature proj (896→512)
+ 12 learnable mask tokens
+ 2-layer Transformer (8 heads)
+ Sine-cosine pos encoding |
| Object Pose Decoder | MLP | [B, 8, 896] | [B, 7] | Feature proj (896→512)
+ 3-layer MLP (512→256→128→7)
+ Quaternion normalization
+ Global avg pooling |
| Mask Decoder | CNN | [B, 8, 896] | [B, 1, 128, 128] | Feature proj (896→512→256)
+ Spatial proj to 8×8×64
+ 4-stage transpose conv upsampling
+ BatchNorm + GELU activation |

Table 5: Decoder specifications

