# OpenReview forum: "AuxVLA: Auxiliary Task Learning and Multi-Modal Enhancement for Vision-Language-Action Models in Mobile Manipulation"
_ICLR.cc/2026/Conference — ICLR 2026 Conference Withdrawn Submission_

### Official Review · Reviewer_exwr · 2025-10-23

**Soundness:** 3
**Presentation:** 3
**Contribution:** 2
**Rating:** 4
**Confidence:** 4

**Summary:**

The paper proposes a pipeline for mobile manipulation. It achieves this through auxiliary task co-training and enhanced input modalities.

**Strengths:**

The paper is clear written. The experiments are thoroughly conducted to validate the effectiveness of each proposed component.

**Weaknesses:**

I think the paper lacks sufficient novelty. It integrates multi-view inputs, which are commonly adopted in many VLA approaches, and adds depth maps that may introduce noise in real-world settings and increase computational cost—one reason why most VLAs avoid using them. The co-training strategy for predicting related actions is also a standard technique frequently seen in prior work.

**Questions:**

1. In line 46, the paper claims that existing methods cannot be applied in dynamic settings. However, since the paper itself does not include any experiments under dynamic conditions, it would be better to remove the word dynamic to avoid overclaiming.

2. In line 192, the term dual-prediction is mentioned, but up to that point, only the flow matching prediction has been introduced. This could be confusing to readers and should be clarified or rephrased.

3. In Section 4.3, the paper uses a ViT for joint prediction and an MLP for end-effector pose prediction. What is the rationale behind this design choice? Some justification or ablation would help.

4. In Table 2, it is unclear what baseline scores are used for comparison—are they taken from Table 1? It would be clearer to present all related results in one place or explicitly reference their source.

---

### Official Review · Reviewer_LvpZ · 2025-10-25

**Soundness:** 3
**Presentation:** 3
**Contribution:** 2
**Rating:** 4
**Confidence:** 3

**Summary:**

This paper proposes a co-training framework for VLA , with a key design of integrating an additional decoder in training phase. This extra decoder is tasked with predicting multiple auxiliary variables related to robot operation, including the robot’s position, joint configuration, grasp state, and object affordance. The core hypothesis of the method is that these auxiliary prediction objectives can implicitly enhance the model’s spatial understanding and physical reasoning capabilities. By aligning the learning process of main VLA tasks with these auxiliary tasks, the paper argues that the model can achieve more robust overall performance on downstream robot control tasks compared to single-task training paradigms.

**Strengths:**

- Comprehensive ablation studies on auxiliary tasks: The paper conducts thorough ablation experiments to evaluate the impact of different auxiliary objectives on the model’s performance. This not only verifies the effectiveness of the proposed co-training strategy but also quantifies the contribution of each individual auxiliary task.

- Novel integration of multi-modal auxiliary signals: The choice of auxiliary variables (robot state + object affordance) is well-motivated, as it bridges low-level physical state information with high-level semantic understanding of objects.

**Weaknesses:**

- Insufficient baseline comparisons: The experiments only include OpenVLA, which is not a state-of-the-art model. Without comparing against state-of-the-art models, it is challenging to assess whether the proposed co-training method can outperform the most advanced approaches or offer unique advantages?
- Limited model scale validation: Given that the Qwen2.5-0.5B model is relatively small, it remains unclear whether the validated conclusions can be generalized to larger-scale models. It remains unclear whether the observed performance gains from co-training can be generalized to larger-scale models (e.g., Qwen2.5-7B). Larger models may exhibit different learning dynamics. For instance, they might already possess stronger spatial reasoning abilities, making auxiliary tasks less impactful, or they could leverage the auxiliary signals more effectively to achieve even greater improvements.
- Lack of real-world evaluation: The study does not include evaluations in real-world scenarios.

**Questions:**

Please refer to the "Weaknesses" section.

---

### Official Review · Reviewer_5pFL · 2025-10-26

**Soundness:** 2
**Presentation:** 2
**Contribution:** 2
**Rating:** 2
**Confidence:** 4

**Summary:**

To address the current poor performance of VLA models in home scenarios and mobile manipulation tasks, as well as their insufficient understanding of dynamic scenes, this paper proposes AuxVLA. Building on the traditional VLA model, this model incorporates multi-view and multimodal inputs to obtain latent space features with stronger representational capabilities. For mobile manipulation tasks, multiple decoders are added to decode information such as chassis position and manipulated object masks to enhance the model's understanding of the physical world. Furthermore, the authors introduce a multimodal training dataset for household mobile manipulation tasks and a three-stage training paradigm (two-stage VLM and one-stage action expert) that maximizes the utilization of the aforementioned model structures. Ultimately, the model significantly outperforms the baseline model on the ManiSkill-HAB benchmark.

**Strengths:**

1. The proposed multimodal input and multimodal prediction mechanisms comprehensively enhance the model's understanding of the physical world, and this capability is crucial for mobile manipulation tasks in home settings.
2. The authors propose a three-stage training paradigm that fully preserves and leverages the inherent reasoning capabilities of the VLM and achieves excellent performance on downstream tasks.
3. The authors thoroughly explore the role of each component of the model and the training paradigm, and identify the optimal state of the art.

**Weaknesses:**

1. There is a lack of real-world experiments in mobile manipulation scenarios. Real-world mobile manipulation scenarios are crucial for validating the performance of the VLA model, and simulator experiments alone make the model's performance less convincing.
2. The baseline selected in the simulator experiments is overly sparse and performs poorly. OpenVLA, as a widely used base model, is no longer state-of-the-art on multiple benchmarks. Therefore, the authors should compare AuxVLA with models such as $\pi_0$. Furthermore, since AuxVLA utilizes deep input and multimodal prediction mechanisms, models using related methods should also be included as baselines.
3. End-to-end mobile manipulation methods often focus on the coordination between chassis movement and manipulator manipulation, such as the AC-DiT mentioned in the paper. However, aside from predicting chassis and manipulator states separately within the multimodal prediction mechanism, AuxVLA does not reflect the coordination between chassis movement and manipulation tasks in the feature space, and experimental results are insufficient to demonstrate this.

**Questions:**

1. Did the authors investigate the performance of AuxVLA on a manipulation-only benchmark compared to a manipulation-only VLA model? Including such experiments would better demonstrate the advantages of the proposed model architecture and training paradigm.
2. The authors include a depth map as input. However, does reusing the RGB encoder to encode the depth map still lose some spatial geometric information? Therefore, is it possible to explore the difference in representational capabilities between the depth map input features and the RGB features, and align the two representations to further improve model performance?

---

### Official Review · Reviewer_K83S · 2025-10-31

**Soundness:** 3
**Presentation:** 2
**Contribution:** 2
**Rating:** 4
**Confidence:** 4

**Summary:**

This paper introduces AuxVLA, a Vision-Language-Action model designed to tackle the challenge of controlling mobile manipulation robots in complex home environments. The core of the method lies in its two-pronged approach: it enriches the model's input with multi-view and depth data for better spatial context, and more importantly, it employs auxiliary task co-training. This involves jointly training decoders to predict a suite of interpretable, intermediate representations (like robot pose, joint states, and object affordances) from shared visual-language features. The key advantage of AuxVLA is its effective solution to controlling a high-dimensional action space, moving beyond simple imitation learning. The authors posit that the auxiliary tasks force the model to develop a superior spatial and scene understanding, which is critical for precise manipulation and navigation, leading to strong performance on a variety of domestic rearrangement tasks and pointing a promising direction for capable domestic robots.

**Strengths:**

1. The research topic of mobile manipulation addressed in this paper remains relatively underexplored, and the tasks involved are highly challenging.
2. This paper proposes an effective training strategy that enables the successful transfer of existing Vision-Language-Action (VLA) models to mobile manipulation tasks.
3. The paper provides comprehensive experimental validation to demonstrate the effectiveness of both the proposed method and its overall design.

**Weaknesses:**

1. The use of multiview and depth inputs is a straightforward and intuitive approach for performance improvement. Presenting this as a core contribution of the paper is not entirely appropriate.
2. To substantiate the claim that the proposed training strategy is more effective than direct imitation learning, the evaluation should extend beyond just the OpenVLA model. Comparisons with other established baselines, such as pi and rdt, are necessary.
3. Additional experiments are required to validate the performance of the model in predicting the final intermediate representations.
4. The dataset used for training is notably large. It would be valuable to explore how to achieve good performance with less data, as this is crucial for real-world deployment where collecting large-scale data is often prohibitively expensive.
5. The submission lacks supplementary videos to visually demonstrate the performance and capabilities of the proposed method.

**Questions:**

please see the weaknesses

**Details Of Ethics Concerns:**

no ethics concerns

---

### Note · Authors · 2025-11-14

I have read and agree with the venue's withdrawal policy on behalf of myself and my co-authors.